# Dendrimeric Structures in the Synthesis of Fine Chemicals

**DOI:** 10.3390/ma14185318

**Published:** 2021-09-15

**Authors:** Bruno Mattia Bizzarri, Angelica Fanelli, Lorenzo Botta, Claudio Zippilli, Silvia Cesarini, Raffaele Saladino

**Affiliations:** Biological and Ecological Sciences Department (DEB), University of Tuscia, 01100 Viterbo, Italy; fanelli.angelica@gmail.com (A.F.); lorenzo.botta@unitus.it (L.B.); zippillic@unitus.it (C.Z.); c.cesarinisilvia@gmail.com (S.C.)

**Keywords:** dendrimers, heterogenous catalysis, materials, PAMAM, nano-devices, anti-virals

## Abstract

Dendrimers are highly branched structures with a defined shape, dimension, and molecular weight. They consist of three major components: the central core, branches, and terminal groups. In recent years, dendrimers have received great attention in medicinal chemistry, diagnostic field, science of materials, electrochemistry, and catalysis. In addition, they are largely applied for the functionalization of biocompatible semiconductors, in gene transfection processes, as well as in the preparation of nano-devices, including heterogeneous catalysts. Here, we describe recent advances in the design and application of dendrimers in catalytic organic and inorganic processes, sustainable and low environmental impact, photosensitive materials, nano-delivery systems, and antiviral agents’ dendrimers.

## 1. Introduction

Dendrimers are symmetric and monodisperse macromolecules with a well-defined three-dimensional branched shape inspired by that of a tree [1]. They show three topologically different regions including: (i) the core part; (ii) the branches formed by repetitive monomeric units (dendrons); and (iii) the “periphery” as final appendages of the dendrons [2]. Two different synthetic methodologies are commonly used for the preparation of dendrimers: (a) the divergent approach, in which dendrimers grow from the core proceeding radially outward toward the periphery; (b) the convergent one, in which the growth starts from the periphery of the dendrimer toward the inner part. The choice of the specific synthetic approach is usually related to the desired application field, depending on both the target chemical structure (sequence of dendrons and order of generation) and the building blocks selected for the construction of the dendritic framework. Dendrimers are classified according to the ‘generation’ level, which represents the minimum number of branching nodes from the core to the periphery. These systems are highly ordered three-dimensional structures often characterized by internal cavities in which metal ions and small molecules can be hosted [3,4].

## 2. Application of Dendrimers in the Science of Materials Field

Dendrimers represent a potent tool in the design of novel sustainable and low environmental impact materials, as optimal reagents, or, alternatively, as structural components in the preparation and functionalization of advanced fibers and composites. For example, they can solve pollution problems associated with the dyeing of cellulose fibers, usually performed with high amounts of electrolyte and alkali solutions [5]. The application of poly-amidoamine (PAMAM) dendrimers modifies the surface of cotton fibers, increasing the dye uptake and fixation without the use of toxic substances [6]. The PAMAM treatment introduces nucleophile amino groups in the polysaccharide able to capture dyes without the use of electrolytes and alkali, minimizing the dye hydrolysis, and reducing the formation of wastes. Dendrimers have also been applied in the design of functional fibers that are active in the removal of organic wastes and bacteria [7]. In this context, PET-1/PAMAM (NH_2_)4-polyester nonwovens [8] showed high catalytic efficacy in the reduction of 4-nitrophenol (4-NP) into 4-aminophenol (4-AP) [9]. In this latter case, the PAMAM terminal amino groups enhanced the adsorption of 4-NP by electrostatic interactions. In addition, the system was applied for the degradation of a large panel of synthetic dyes, including Malachite Green (MG), Methylene Blue (MB), and Reactive Red 195 (RR), using NaBH4 as reducing agent. As an improvement, a three-layer supramolecular system based on the sandwich between PET-1/Dr and polyester nonwovens functionalized with 3-(aminopropyl)trethoxysilane (APTES) and copper and silver atoms have been successfully applied as anti-bacterial materials. The functionalization of dendrimers with chromophore groups offers the possibility to create photosensitive materials to be used in catalysis, sensing, and biomedicine [10,11]. For example, photochromic dendrimers synthetized through Michael addition reaction between poly(amidoamine) dendrimers with three different generations (G1, G3, and G5) and azo-benzene acrylate (Azo), denoted as Gn-Azo (n = 1, 3 and 5), have been applied in the preparation of photo-switchable adhesives and healable coatings (Figure 1) [12].

In this system, the azo-benzene group is subject to photo-isomerization phenomena (trans versus cis isomer) associated to the n-π* transition after UV irradiation. Conversely, visible light irradiation reverted the isomerization to trans form [13,14]. This effect influences the glass transition temperature of dendrimers [15], making them available for reversible adhesive capability when irradiated with visible or UV light. Interestingly, this property was found to increase by raising the dendrimer generation. For example, the adhesion strength of the fifth-generation dendrimers (G5-Azo) can reach up to 1.62 MPa, which is five times higher than that of pristine azo-benzenes. Finally, the use of dendrimers in the production of controlled porous silica materials for catalysis, bio-separation, and biomedicine applications, has been reported and discussed [16]. In this context, PAMAM characterized by a tetra-functional ethylene-diamine core acts as an organic template for the preparation of mesoporous silica materials [17]. PAMAM generation zero to three bearing amine terminal group and PAMAM generation two bearing alkylated quaternary ammonium terminal groups produced mesoporous silica materials with high mono-dispersion of the pre diameter (3–9 nm). In addition, due to the hydrogen bond affinity between water and amino group at the periphery of the PAMAM, the template was easily removed from the silica by a simple extraction procedure, without the use of polluting organic solvents. PAMAM recycling tests showed that the dendrimer is rarely wasted and maintains good chemical stability, furnishing well-defined mesoporous silica material with large surface area, pore volume, controllable pore diameter, and narrow pore size distribution.

## 3. Nano-Delivery Systems

The well-defined three-dimensional structure of dendrimers make them a suitable class of macromolecular nano-scale delivery devices [18] with promising solubility and targeted delivery properties [19]. Dendrimer nano-devices conjugated with gallic acid have been reported as a novel drug-delivery strategy for the therapy of chemo-resistance in neuroblastoma cells [20]. Human neuroblastoma (NB) is a pediatric tumor that usually develops resistance to therapy due to the overproduction of reactive radical oxygen species (ROS). In this regard, gallic acid can prevent cancer due to the presence of the phenolic pharmacophore [21,22]. The bioavailability of gallic acid is low due to its low solubility in both water and alcohols [23]. Biodegradable dendrimer based on a fifth-generation polyester (G5) inner matrix with tree-like repetition of 2,2-bis (hydroxymethyl)propionic acid and 64 peripheral hydroxyl groups, namely matrix 4 (Figure 1), has been applied as a gallic acid-enriched nano-device for the treatment of chemo-resistant neuroblastoma cells [24].

Gallic acid, a polyphenol largely used beside its derivatives as a potent antioxidant [25,26,27], was supported by a covalent bond with the dendrimer backbone, or, alternatively, by encapsulation and surface absorption on the polymer matrix 4 (Figure 2).

Functionalized dendrimers exhibited a ROS-mediated cytotoxic activity on chemo- resistant cancer cells at a concentration of almost 100 times lower than that of gallic acid alone. The conjugation strategy, involving dendrimers and bioactive substances, was also effective in the case of PAMAM dendrimer-azithromycin conjugate nano-devices active against chlamydia trachomatis infections [28]. Chlamydia trachomatis is an obligate intracellular bacterium responsible for sexually transmitted infections that are resistant to antibiotic therapy [29,30]. Azithromycin (AZ)-conjugated 4-generation (G4) of PAMAM dendrimers bearing OH group in the terminal position (PAMAM G4-OH-AZ) have been synthesized in order to use the hydroxyl-end-functionality as “ester-link” for the controlled release of the drug (Figure 2).

Azithromycin was first reacted with glutaric acid using DCC, DMAP, and trimethylamine, followed by esterification with PAMAM G4-OH. Kinetic studies of the release of AZ confirmed the efficacy of the nano-device which was more effective than AZ alone in the treatment of Chlamydia Trachomatis. A further study described the use of a third generation (G3) of PAMAM dendrimers as nano-carriers of paclitaxel, a known anticancer agent inhibitor of P-glycoprotein and Topoisomerase 1 [31,32]. Paclitaxel is applied in the treatment of various cancers, including non-small cell carcinomas of the ovary, breast, and lung [33]. However, its therapeutic efficacy is limited due to poor solubility and permeability. To solve these biases, paclitaxel (Pac) has been covalently linked via a glutaric anhydride (glu linker) to PAMAM dendrimer bearing lauryl acid side-chains (lauryl-G3 PAMAM: G3L3, G3L6) [34]. In this way, Pac was linked to glutaric anhydride via an ester bond and to lauryl-G3 PAMAM or G3 PAMAM via an amide bond (Figure 3) [35].

Modified dendrimers showed appreciable activity against human colon adenocarcinoma cell line (Caco-2) and primary cultured porcine brain endothelial cells (PBECs). The analysis showed an increased toxicity of the modified dendrimers with respect to the native counterpart. In addition, the novel nano-devices showed the highest permeability in both apical-to-basolateral and basolateral-to-apical directions across both cell monolayers, probably, due to the occurrence of a lauryl side-chain favored interaction with the plasma membrane. The surface engineering of dendrimers can further improve their application as nano-carriers. For example, the surface of G4 PAMAM dendrimers have been modified by using polyethylene glycol and folic acid [36] in order to reduce the amount of free terminal NH_2_ groups which are responsible for toxicity [37,38,39]. The application of these platforms for the selective release of anti-cancer bioactive agents, such as 5-fluorouracil (5FU), was reported focusing on the effect against human colorectal cancer (Figure 3).

Due to their biodegradability and biocompatibility profile, dendrimers have also been used in agricultural and agro-industrial fields. For example, water-soluble nano-sized cationic dendrimers containing hydrophobic and peripheral dendritic polyester amines demonstrated their capability to carry out the hydrophobic pesticide thiamethoxan [40]. In this latter case, three generations (G1, G2, and G3) of functionalized cationic dendrimer-based nano-carriers, consisting of a fluorescent perylene-3,4,9,10-tetracarboxydiimide chromophore core (PDI) and amino groups at the periphery (Figure 4), were used as a nano-device by loading of the pesticide through formation of electrostatic interactions [41]. The drug release was significantly facilitated by the terminal amino groups that increased the water solubility of the dendrimer. All three generations of water-soluble fluorescent PDI-cored cationic dendrimers effectively delivered the drug in live cells; in particular, second and third generations of nano-carriers showed an increasing drug cytotoxicity to insect pests with respect to the drug alone.

## 4. Application of Dendrimers in Catalysis

Dendrimers are efficient platforms for the preparation of heterogeneous catalysts based on metal nanoparticles, such as palladium nanoparticles. Pd-based dendrimers have been reported as a green alternative for palladium-based organometallic reactions, including Still, Sonogashira, Mizorocki–Heck, and Suzuki–Miayura reactions [42,43,44]. Catalytic systems encompassing palladium nanoparticles immobilized on dendrimers showed simple recoverability and reusability, avoiding drawbacks related to the leaching of the metal, purification of final products, and toxicity of common metal ligands. In addition, the combination of magnetic nanoparticles and dendrimers opened a new entry for fast, simple, and inexpensive recovery procedures [45,46,47]. In this context, capped guanidine third-generation dendrimer supported on Fe_3_O_4_@SiO_2_ nano-materials were used for the encapsulation of Pd nanoparticles (G3-Gu-Pd; Figure 4) [48], and applied in the Mizoroki–Heck and copper-free Sonogoshira reactions under environmentally friendly conditions. The authors reported several examples of the Mizoroki–Heck coupling reaction between aryl halides 1 and alkenes 2 in the presence of G3-Gu-Pd catalyst (K_2_CO_3_, SDS surfactant) in water at 100 °C to yield 3 in high yields (62–91%). In addition, G3-Gu-Pd catalyst was applied in the copper-free Sonogashira reaction of aryl halides 4 with acetylene compounds 5 to afford 6 in high yields (60–90%).

The use of silica supported phosphine–palladium terminated PAMAM dendrimers in the Heck condensation between aryl bromides and a panel of alkenes, such as styrene and butyl acrylate, was also reported (Figure 5) [49]. The catalyst was prepared by reaction of palladium complex [Me_2_NCH_2_-CH_2_NMe_2_]PdMe_2_ and the supported phosphine dendrimer (G0–G4), with the stability of the system being increased significantly from G0 to G4 generation of PAMAM. These catalysts were effective in the coupling between bromo-benzene and styrene to yield (E)-stilbene with appreciable selectivity, opening new synthetic perspectives preparation of bioactive stilbene derivatives [50].

Highly efficient heterogeneous catalysts for the hydrogenation of conjugate double bonds have been prepared by combining silica–polyamine composite (SPC BP-1) with poly-propyleneimine (PPI) dendrimers containing Pd nanoparticles (Figure 5). These catalysts were effective for the selective hydrogenation of isoprene, phenyl-acetylene, and 2,5-dimethyl-2,4-hexadiene [51,52,53]. The catalyst was obtained by reacting the imine derivative of SPC BP-1 with the third-generation of poly-propyleneimine (PPI) dendrimers, followed by loading with Pd(II) as Pd(OAc)_2_. The reduction of the Pd(II) precursor with sodium borohydride afforded the effective Pd(0) DAB(NH_2_)16*Pd^0^@BP1 (Figure 5). This hybrid showed high efficiency in the hydrogenation of dienes under hydrogen pressure of 3 MPa and a substrate/Pd ratio of 66 240, resulting in 85% of conversion and a selectivity of alkene products of 98%. No leaching of Pd was reported and the catalyst was effective for four cycles without loss of activity. Moreover, the catalyst showed higher activity in comparison to PPI dendrimers alone or Lindler catalyst.

Dendritic heterogeneous catalyst based on palladium nanoparticles have also been reported to effectively catalyze C-S cross-coupling reactions. In this context, palladium nanoparticles immobilized on nano-silica triazine dendritic polymer (Pdnp-nSTDP) are effective catalysts for the synthesis of diaryl and aryl-heteroaryl sulfides under microwave (MW) irradiation (230 W, 80 °C), in the presence of tetra-butylammonium hydroxide (TBAH) in 1 H_2_O-DMF mixture (Figure 6) [54]. Pdnp-nSTDP were prepared starting from a nano-silica 3-amino-propyl-trimethoxysilane platform, followed by the in situ generation and entrapment of Na_2_Pd_2_Cl_6_. Pdnp-nSTDP were also effective in the C-S cross coupling with low reactive aryl chlorides (Figure 6). The catalyst retained its activity for five successive runs.

The ability of dendrimers to adsorb and stabilize metal nanoparticles inspired their application in the environmental remediation of toxic metals, such as copper, lead, and chromium, in the aqueous environment. For example, poly(N-propylethane-1,2-diamine) G3 bearing a magnetite core and capped with a carboxylic acid moiety (Fe_3_O_4_/SiO_2_/PNPEDA G3-COOH) showed a high efficacy in the chelation of heavy metals (Figure 6) [55]. In addition, it was easily recoverable and regenerated by a simple acid treatment.

## 5. Dendrimers Application in Organo-Catalysis

The application of PAMAM dendrimers in different fields of organic chemistry, including multicomponent reactions, heterocycles synthesis, and organo-catalysis, are reported and discussed [56]. Porphyrin-cored poly(amidoamine) (G1 POR-PAMAM) dendrimers act as homogeneous catalysts in the multi-component Biginelli and Hantzsch reactions [57]. G1 POR-PAMAM were synthesized by a multi-step procedure encompassing the nucleophilic ring opening of epichlorohydrin with hydroxyphenyl porphyrin, azidation, and reduction, followed by methyl acrylate tandem Michael addition and trans-amination with ethylenediamine (Figure 7). G1 POR-PAMAMs catalyzed the synthesis of 1,4-dihydropyridine and 3,4-dihydropyrimidin-2(1H)-thione derivatives, starting from aldehydes, ethyl-acetoacetate, and ammonium thiocyanate, in a single step and in a short time (45 min), with almost quantitative yield (92–99%) (Figure 7).

A similar procedure has been applied for the preparation of pentaerythritol-based dendritic nanostructures endowed of catalytic activity [58]. In this latter case, the authors applied an azide-mediated ring opening of epichlorohydrin, followed by reduction of the cyano moiety, tandem Michael addition, and trans-amination with ethylene diamine. The growing of the dendrimer was characterized by the use of MALDI-TOF/MS, then FTIR, NMR, and, when necessary, by TEM microscopy.

The catalytic activity of these PAMAMs was evaluated on the multicomponent synthesis of bis-imidaziole derivatives starting from 1,2-diketones, aromatic aldehydes, bis(3-aminopropyl)amine, aromatic aldehydes, and ammonium acetate (Figure 8). Electron-donating groups (-OH, -OCH_3_) on the aromatic aldehyde increased the yield of the products compared to electron-withdrawing substituents (-NO_2_, -Cl), with the catalyst retaining its activity for more runs. Some of the novel compounds showed inhibitory activity against 17β-hydroxysteroid dehydrogenase type 1 (17β-HSD1, PDB code 3HB5), a key enzyme active in breast cancer and estrogen-related diseases.

Dendronized polymers in which cavity and surface functionalities are the reactive centers of unimolecular micelles self-assembled in aqueous medium have been prepared from tris(hydroxymethyl)propane, epichlorohydrin, sodium azide, and acrylonitrile (instead of methyl acrylate) [59]. These aggregates act as homogeneous organo-catalysts for the one-pot synthesis of 4-aryl-1H-1,2,3-triazoles, starting from aldehyde, nitromethane, and sodium azide (Figure 9). It was observed that *meta*-substituted aldehydes afforded lower yields compared to *ortho*- and *para*- substituted aldehydes, and the formation of dendritic unimolecular micelle in aqueous solution simplified the product isolation and catalyst recycling. The product could be isolated by simple extraction with organic solvents, and the dendritic molecular micelle was recovered by simple centrifugation of the water medium. In addition, the catalyst was reused up to six cycles retaining a high activity.

The role of PAMAM functionalized with chloro-sulfonic acid as Brønsted acid organo-catalyst has been reported [60]. In this latter case, ethylene diamine was polymerized with methyl acrylate and chloro-sulfonic acid to generate the sulfamic acid functionalities (Figure 10). The novel catalyst was used for the synthesis of acetals from aldehydes and acetic anhydride at room temperature and under solvent-free conditions. In addition, the catalyst was able to perform the inverse reaction of de-protection of acetals to corresponding aldehydes, showing, in both cases, a good recyclability.

The mechanism of action was rationalized by the study of the Hammett acidity function and compared with NH_2_SO_3_H. A strong Brønsted acid behavior was observed as a consequence of the multi-functional and high proton loading level of the dendrimer (Figure 11).

PAMAM dendrimers can also perform as a coordinative functional platform for the immobilization of reactive heavy-metal/benzo-15-crown-5 ether complexes on multi-wall carbon nanotubes (MWCNTs) [61]. Three different heterogeneous catalysts were obtained by the following procedures: (i) introduction of ethylene diamine spacer on inert MWCNTs and decoration of the amine functionalities with benzo-15-crown-5 ether to yield linear dendrimer species (Figure 12); (ii) dendrimerization of the diamine spacer on the carbonaceous nanotube by methyl acrylate addition followed by crown ether coupling (Figure 13).

In both cases, the catalysts were loaded with MTO that is a potent activator of hydrogen peroxide by formation of rhenium peroxide intermediates (Figure 14). The novel branched catalysts efficiently transformed olefins into the corresponding epoxides, while the linear counterpart afforded mainly diol derivatives.

## 6. Dendrimers as Antiviral Agents

Peptides and derivatives have shown an enormous therapeutic potential for the treatment of different viral infections, due to their high biocompatibility, selectivity for targets, low toxicity, and easy elimination [62,63]. The effectiveness of peptides in the inhibition of viral agents has been improved by the synthesis of dendrimers. The antiviral efficacy of dendrimers is usually associated to their similarity to biomimetic peptides, high stability (protease resistance), lack of conformational effects, and, when necessary, high capability to carry out drugs [64]. In this latter case, the hydrophobic drug is bonded near the hydrophobic cores, while the hydrophilic external surface favors the solubility of water. Dendrimers showed appreciable antiviral activity against influenza virus, human immunodeficiency virus, and respiratory syncytial virus. 

The functional groups located at the periphery of the system can block the entry of the virus by competing with the cell, (Figure 8) or by exerting direct effects on viral replication [65,66]. In particular, dendrimers modified with naphthyl residues and sulfonate groups have been found to be active against Herpes Simplex virus and HIV [67]. Peptides conjugated with dendrimers exhibit several benefits with respect to corresponding peptides or dendrimers alone. Peptide dendrimers are easily synthesized, preserve their activity at low- and high-salt conditions, are resistant to the activity of proteases, and generally show a low toxicity. Sialic acid conjugated poly-amidoamine (PAMAM) dendrimers (G4-SA) have been synthesized in the frame of the prevention of influenza A H1N1, H2N2, and H3N2 infection. These compounds showed inhibitory activity against H3N2 and H1N1 influenza subtypes in hemagglutination-inhibition assays. Other types of dendrimers bearing a charged group on the surface have also been reported as inhibitors of MERS-CoV. These systems are anionic dendritic polymers, including hydroxyl, carboxyl, and succinic acid substituents, or in alternative cationic dendritic polymers containing primary amine end groups.

In particular, polymers bearing a carboxyl and a succinic acid moieties exhibited the best inhibition values, probably, due to a hydrogen and p–p stacking interaction between the viral fusion protein and the dendrimer [68]. PAMAM dendrimers have also been conjugated with different sugar moieties as potential inhibitors of human and avian influenza virus strains. In this latter case, sialyllactose (6SL) PAMAM derivatives showed interesting activity against human and avian strains. The application of sialyllactose moiety in the PAMAM dendrimers was further analyzed in the case of PAMAM dendrimers of generation 4 and 5 bearing up to 128 sialyllactose ligands. 

These compounds showed a potent antiviral activity in the low micro-molar range [69,70]. In addition, G1–G3 poly-anionic carbosilane dendrimers (Figure 9) have recently been reported as active compounds against HIV-1 [71]. When tested alone or in combination with latency-reversing agents (LRAs) bryostatin-1,2, panobinostat, and romidepsin (RMD), they significantly increased the expression of cell lines latently infected with HIV-1 p89GFP (GFP). The results showed an enhanced expression of GFP until 80%, proving that a combinatory treatment in the “shock and kill” method with dendrimers could also help with the reactivation activity of LRAs.

## 7. Conclusions

Dendrimers are useful candidates for a large number of applications, including the design of sustainable and low environmental impact advanced materials, catalytic processes, nano-delivery systems, and applications in the pharmacological field. Although they were studied for many years, the synthesis of these compounds still requires multi-step chemical reactions, which encompass two main categories: (a) the convergent approach; (b) the divergent approach [72]. These synthetic procedures can afford dendrimers with a controlled size and shape, finely tuning the properties of surface, branches, and core. In this context, solubility, reactivity, and biocompatibility can be modulated easier than polymers with a lower degree of structural order and repetitive motif [73]. The design of dendrimers is strictly related to their function requiring particular attention for medical applications [74]. Dendrimers have a well-defined nano-sized structure which makes them appropriate for oral, parenteral, pulmonary, and nasal drug delivery, due to their ability to cross the cell membrane by both transcellular and paracellular pathways. On the other hand, their toxicity is still a matter of debate [75,76]. Their main drawback relates to the fate of their interactions with biological membranes, which can in principle afford the disruption of the membrane and cell death [77]. In order to overcome these limitations and avoid undesirable side effects, several approaches emerged mainly based on the surface modifications strategy [73]. Although further studies are still required in order to fully characterize the safety profile of dendrimers, they possess unique delivery properties for both hydrophilic and lipophilic drugs, and some of them have received FDA approval, or, alternatively, are in the early stages of clinical trials [78].

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
