# Peer review of "Dendrimeric Structures in the Synthesis of Fine Chemicals"

_materials, 2021, doi:10.3390/ma14185318_

Round 1
Reviewer 1 Report
Summary: Reconsider after major revision.
The authors have presented a review of dendrimers and their synthesis in the context of various applications.
- A small section or table on the advantages and disadvantages of dendrimers over other polymeric forms/delivery systems for each section is needed.
- The introduction is very bare-bones and needs to be expanded.
- This manuscript would benefit from more references (currently at 71), especially more recent references (only 18 are 2017 or newer).
Author Response
We thank the reviewer for all the suggestions.
Q1. A small section or table on the advantages and disadvantages of dendrimers over other polymeric forms/delivery systems for each section is needed.
A1. We agree with the referee suggestions. A “conclusion” section has been added in the manuscript including a general overview on the advantages and disadvantages of dendrimers over other polymeric forms and delivery systems.
Q2. The introduction is very bare-bones and needs to be expanded.
A2. The introduction section has been improved in the revised version of the manuscript including the description of the main procedures for their synthesis.
Q3. This manuscript would benefit from more references (currently at 71), especially more recent references (only 18 are 2017 or newer).
A3. New recent references have been added in the manuscript.
Reviewer 2 Report
This manuscript reviewed the recent progresses in the application of dendrimers in drug deliveries and chemical synthesis. I would suggest this manuscript be accepted by Materials with minor revision.
- The well-defined three dimensional structure of dendrimers make them as a promising nanoscale delivery tool for many medicines. The application of dendrimers as drug delivery tool was discussed in part 3. I would suggest adding a brief discussion on the recent toxicity study for dendrimers.
- Are there any laws to follow about the design of the dendrimer structure for different applications (such as the core, the surface modification, or the generations)?
Author Response
We thank the reviewer for all the suggestions.
Q1. The well-defined three dimensional structure of dendrimers make them as a promising nanoscale delivery tool for many medicines. The application of dendrimers as drug delivery tool was discussed in part 3. I would suggest adding a brief discussion on the recent toxicity study for dendrimers.
A1. A discussion concerning the recent toxicity study for dendrimers has been added in the conclusion section of the manuscript.
Q2. Are there any laws to follow about the design of the dendrimer structure for different applications (such as the core, the surface modification, or the generations)?
A2. A brief overview concerning the main design strategies and synthetic methodologies for planning the dendrimer structures in now inserted in the introduction and in the conclusion sections of the manuscript.
Reviewer 3 Report
Dear Editor,
The review manuscript submitted by Bruno M.Bizzarri et al. titled “Dendrimeric structures in the synthesis of fine chemicals.” described recent advances in the design and application of dendrimers in the field of materials science, nano-delivery systems, application in catalysis, organo-catalysis and antiviral agents.
It is of high quality to be published in the journal of Materials.
Regards,
Supplement:
Dear Editor,
The review manuscript submitted by Bruno M.Bizzarri et al. titled “Dendrimeric structures in the synthesis of fine chemicals.” described details of recent advances in the design and application of dendrimers in the field of materials science, nano-delivery systems, application in catalysis, organo-catalysis and antiviral agents.
The following minor points need to be revised before this work can be published:
Minor Comments:
1. In the abstract, author mentioned that “Here we describe recent advances in the design and application of dendrimers in catalytic organic and inorganic processes encompassing the synthesis of fine chemicals, drugs and novel functional materials.”
I recommend to modify the abstract, based on the detailed topics in this review including sustainable and low environmental impact, photosensitive materials, nano-delivery systems, and antiviral agents’ dendrimers.
2. It is not clear what the matrix 4 in Figure 1 & 2 is.
3. In Figure 2, the core sphere shape on the centre of the dendrimer structure is unmarked, what is it?
4. I recommend that authors use a consistent terminology for the different generation of dendrimers throughout the manuscript in the text and figures. For example in Scheme 2, “4-generation PAMAM dendrimers” was used and in figure 3 “G4 dendrimers”.
5. Figure 3, page 6, change to Figure 4.
6. Scheme 3, should be after the text “Paclitaxel is applied in the …and to lauryl-G3 PAMAM or G3 PAMAM via an amide bond (Scheme 3) [35].”
7. I recommend that a brief conclusion/vision for the future to be added to this review.
Author Response
We thank the reviewer for the positive comments.
Q1. In the abstract, author mentioned that “Here we describe recent advances in the design and application of dendrimers in catalytic organic and inorganic processes encompassing the synthesis of fine chemicals, drugs and novel functional materials.”I recommend to modify the abstract, based on the detailed topics in this review including sustainable and low environmental impact, photosensitive materials, nano-delivery systems, and antiviral agents’ dendrimers.
A1. The abstract has been modified as suggested.
Q2. It is not clear what the matrix 4 in Figure 1 & 2 is.
A2. We agree with the referee consideration. The matrix is referred to a fifth-generation polymer, this aspect is now clarified by adding the acronym “(G5)”, representing the order of the generation of the polymer, in the appropriate section of the text.
Q3. In Figure 2, the core sphere shape on the centre of the dendrimer structure is unmarked, what is it?
A3. The core sphere shape of Figure 2 is referred to the polymer matrix 4. This point is now clarified both in the text and in the caption of the Figure.
Q4. I recommend that authors use a consistent terminology for the different generation of dendrimers throughout the manuscript in the text and figures. For example in Scheme 2, “4-generation PAMAM dendrimers” was used and in figure 3 “G4 dendrimers”.
A4. The manuscript has been revised accordingly.
Q5. Figure 3, page 6, change to Figure 4.
A5. Ok Done
Q6. Scheme 3, should be after the text “Paclitaxel is applied in the …and to lauryl-G3 PAMAM or G3 PAMAM via an amide bond (Scheme 3) [35].”
A6. Ok Done
Q7. I recommend that a brief conclusion/vision for the future to be added to this review.
A7. A conclusion section has been added in the revised version of the review.
Reviewer 4 Report
A review by Bizzarri et al focuses dendrimers in synthesis of fine chemicals. The manuscript is well written and structured. The figures are clear and well prepared.
Comments:
- Is it necessary to split figures into figures and schemes?
- "Figure 3" name is duplicated.
- The manuscript lacks conclusions/summary.
Author Response
We thank the reviewer for the comments.
Q1. Is it necessary to split figures into figures and schemes?
A1. In our opinion Figures and Schemes are useful for a better reading of the manuscript, the schemes being mainly devoted to the description of synthetic pathways.
Q2. "Figure 3" name is duplicated.
A2. Thanks, we have modified accordingly
Q.3 The manuscript lacks conclusions/summary.
A3. A conclusion section has been added in the revised version of the manuscript.